

# Publication delays and associated factors in ophthalmology journals

Yinglin Yu[1,*], Wei Li[2,*], Chaoqun Xu[1], Yuan Tan[1], Weining Zhu[2], Bowen Zhang[2], Yingshi Zou[1], Leyi Hu[1], Guangming Jin[1] and Zhenzhen Liu[1]

[1] State Key Laboratory of Ophthalmology, Zhongshan Ophthalmic Center, Sun Yat-sen University, Guangdong Provincial Key Laboratory of Ophthalmology and Visual Science, Guangdong Provincial Clinical Research Center for Ocular Diseases, Guangzhou, Guangdong, China
[2] Sun Yat-sen University, Zhongshan Medical School, Guangzhou, Guangdong, China
* These authors contributed equally to this work.

## ABSTRACT

**Introduction:** This study aimed to evaluate the publication delays and correlative factors of peer-reviewed ophthalmology journals.

**Methods:** The ophthalmology journals listed in the Journal Citation Report 2020 were retrieved from the Web of Science database. The first original research article of each journal issue from January to December 2020 was extracted, and its submission, final revision, acceptance, and publication dates were obtained. Information on impact factors, advance online publication (AOP) status, open access (OA) rate and acceptance rate in 2020 was also collected. The correlations between publication delays and potential associated factors were analyzed.

**Results:** A total of 58 ophthalmology journals were included and information on 685 articles was collected. The median times from submission to acceptance, from acceptance to publication, and from submission to publication were 118.0 (IQR, 74.0–185.0) days, 31.0 (IQR, 15.0–64.0) days, and 161.0 (IQR, 111.0–232.0) days, respectively. A higher impact factor was correlated with shorter delays of acceptance and publication ($P < 0.05$). There was a positive correlation between acceptance rates and publication delays ($r = 0.726$, $P = 0.007$). Forty-seven (81.03%) journals provided AOP. There was no statistically significant difference for impact factors and publication delays between journal with and without AOP (all $P > 0.05$). No correlation between OA rate and publication delays or impact factors was detected (all $P > 0.05$).

**Conclusions:** Journals with higher impact factors and lower acceptance rates tend to have quicker publication processes. No significant associations were detected between publication delays and AOP or OA rate.

# INTRODUCTION

Scholarly communication in science, technology, and medicine has always been organized around journal publishing (*Baffy et al., 2020*). The publication speed of research papers affects the timeliness of the dissemination of scientific knowledge, as well as researchers'

Corresponding authors
Guangming Jin,
jingm@mail2.sysu.edu.cn
Zhenzhen Liu,
liuzhenzhen@gzzoc.com

academic influence, and even careers. Publication times determine the dissemination of important findings, which further affects their contribution to scientific progress. Timely publication may promote positive communication at academic conferences (*Vosshall, 2012*), provide the latest evidence for secondary research (*Yu, Rong & Li, 2003*), and enable patients to benefit from treatments based on the best evidence (*Shephard, 1973*) and decision makers to make reasonable and beneficial decisions (*Christie et al., 2021*). In addition to promoting scientific developments, the publication of research also bears witness to researchers' contributions. Publication speed may play an important role in researchers' promotion and access to funding and advanced platforms (*Vosshall, 2012*). Research on the publication speed and impact factors of ophthalmology journals is helpful for journal selecting and provide useful information for global scholars.

In recent years, research on factors affecting journal publication speed and impact factors has attracted more and more attention (*Jain et al., 2021*; *Kalcioglu et al., 2015*; *Mohanty et al., 2021*; *Shah, Sherighar & Bhat, 2016*). An increasing number of journals have adopted advance online publication (AOP) and open access (OA) models to accelerate the publication of papers and remove barriers to access to scientific knowledge (*Nature Publishing Group, 2002*; *Björk & Solomon, 2013*). The AOP and OA models have improved the efficiency and breadth of the dissemination of scientific achievements with benefit to both authors and readers (*Nature Publishing Group, 2002*; *Björk & Solomon, 2013*; *Laakso & Björk, 2012*). However, OA journals or options requiring an article-processing charge to fund publishing can be a barrier for authors and institutions without fundings to publish in an OA format (*Björk & Solomon, 2012*).

Little research has been conducted on the impact of these models on publication times and impact factors. Few studies have investigated the publication times and associated factors of ophthalmology journals (*Chen, Chen & Jhanji, 2013*; *Dhoot et al., 2021*; *Skrzypczak et al., 2021*). The most recent systematic evaluation of publication times in ophthalmology journals and the influence of AOP on impact factors was conducted by *Chen, Chen & Jhanji (2013)* concerned the year 2010. These studies have significant limitations, such as being outdated (*Chen, Chen & Jhanji, 2013*), with small sample sizes (*Skrzypczak et al., 2021*), and the fact that they did not explore the relationships between OA status and publication delays or impact factors (*Chen, Chen & Jhanji, 2013*; *Dhoot et al., 2021*; *Skrzypczak et al., 2021*).

Therefore, this study aims to evaluate the publication delays of ophthalmology journals in 2020 and the relationships between publication delays, impact factors, AOP, OA rate, and acceptance rate, hoping to help authors select appropriate journals for article submission.

## METHODS

### Data collection

The ophthalmology journals listed in the Journal Citation Report 2020 were retrieved from the Web of Science (https://jcr.clarivate.com/jcr/browse-journals; accessed October 4, 2021). Review-only journals were excluded. Only original research articles were included. Review articles, editorials, correspondence, meta-analysis and case reports were excluded.
The first article of each issue or each month between January and December 2020 was selected as representative of each journal. Supplementary issues were excluded. For journals with a certain number of issues published per year like *Ophthalmology*, we chose the first article published in each issue as representative articles of the journal. For journals which did not publish a certain number of issues per year, the first article published each month was chosen as representative. Twelve articles were eventually included in each journal. Information on the submission, final revision, acceptance, and publication dates was obtained from the full text of each article published in print or online only. Submission date indicates the time when an article was submitted for consideration for publication in a journal. Acceptance date indicates the time of communication of final decision to the corresponding author of an article. AOP date means the time of publication of the article online in advance while print publication date means the time when the article was actually printed or the date in which final pagination or bibliometric details were added to the article. The data were extracted independently by two investigators. Any discrepancies between the two investigators were resolved through discussion with a third investigator. The interval times from submission to acceptance (SA), from acceptance to online publication of journals with AOP or print publication of that without AOP (AP), and from submission to online publication of journals with AOP or print publication of that without AOP (SP) were calculated for each article. As for journals with AOP, the interval between acceptance and print publication (APP) was also measured. Information on the OA rate of each journal was obtained from the Journal Citation Report 2020 retrieved from the Web of Science and the acceptance rate was obtained from the webpage of each journal. The study was conformed to the tenets of the Declaration of Helsinki and performed under institutional review board approval.

## Statistical analysis

The medians and interquartile ranges (IQRs; 25–75%) of SA, AP and SP were calculated. The Wilcoxon test or the Mann–Whitney $U$ test was used to assess differences in APs and APPs between journals with and without AOPs and to compare the impact factors of journals with and without AOPs. Spearman's rank correlation coefficient was used to evaluate the correlation between the impact factor, OA rate, acceptance rate and publication times. The Shapiro-Wilk test was used to test the normality of variables and non-parametric analysis was performed for data that were not normally distributed. $P < 0.05$ was considered to be statistically significant. R software version 3.6.1 (R Development Core Team, Vienna, Austria) was used for the statistical analysis.

## RESULTS

A total of 58 journals were included in the analysis. Among them, seven (12.1%), 35 (60.3%), and six (10.3%) journals did not indicate submission, revision, and acceptance dates, respectively (Table S1).

Table 1 lists the median peer review and publication delays of journals with necessary information. The individual median peer review times (SA) ranged from 49.0 to 251.5

**Table 1 Peer review and publication time lag of ophthalmology journals in 2020.**

| Journal | Median time between submission and acceptance (interquartile range), days | Median time between acceptance and publication (interquartile range), days | Median time between submission and publication (interquartile range), days | Impact factor | No. of issues/ year | No. of articles examined |
|---|---|---|---|---|---|---|
| Acta Ophthalmol | 79.0 (66.0–212.5) | 27.5 (25.5–30.5) | 111.5 (96.5–237.5) | 3.376 | 8 | 12 |
| Am J Ophthalmol | | 10.5 (8.0–13.5) | | 5.258 | 12 | 12 |
| Arq Bras Ofthalmol | 183.5 (109.5–359.5) | 322.0 (265.0–417.5) | 594.5 (413.5–701.5) | 0.900 | 6 | 12 |
| Asia-Pac Journal of Ophthalmol | 77.5 (62.5–113.5) | 63.0 (45.0–113.5) | 159.5 (137.0–193.5) | 2.827 | 4 | 12 |
| Br J Ophthalmol | 94.5 (71.0–127.5) | 18.5 (13.0–22.0) | 111.5 (87.0–146.5) | 4.638 | 12 | 12 |
| BMC Ophthalmol | 177.0 (117.0–301.0) | 13.0 (8.0–21.5) | 182.0 (138.5–321.5) | 2.209 | | 12 |
| Can J Ophthalmol | 73.0 (44.0–187.0) | 70.0 (46.5–82.5) | 143.5 (119.5–253.5) | 1.882 | 6 | 12 |
| Clin Exp Ophthalmol | 137.0 (101.5–190.0) | 23.0 (15.0–25.5) | 174.5 (120.5–213.0) | 4.207 | 8 | 12 |
| Clin Exp Optom | 108.5 (63.0–158.5) | 48.5 (35.0–51.0) | 154.0 (108.5–211.0) | 2.742 | 6 | 12 |
| Contact Lens Anterior Eye | 155.5 (82.0–271.5) | 11.0 (7.0–17.0) | 165.5 (95.0–287.0) | 3.077 | 6 | 12 |
| Cornea | 178.0 (93.0–225.0) | 62.0 (43.5–89.0) | 265.5 (165.5–296.5) | 2.651 | 12 | 12 |
| Curr Eye Res | 108.0 (65.0–144.0) | 23.5 (21.5–28.5) | 139.0 (86.0–223.0) | 2.424 | 11 | 12 |
| Cutan Ocul Toxicol | 71.5 (54.5–126.0) | 23.0 (20.5–37.0) | 121.0 (78.5–195.5) | 1.820 | 4 | 12 |
| Doc Ophthalmol | 141.5 (98.0–167.0) | 11.5 (7.5–17.5) | 151.5 (107.5–181.0) | 2.379 | 6 | 12 |
| Eur J Ophthalmol | 160.5 (90.5–248.0) | 29.5 (21.5–65.0) | 226.5 (154.0–302.0) | 2.597 | 6 | 12 |
| Exp Eye Res | 125.0 (100.0–156.5) | 6.5 (2.5–8.5) | 139.5 (106.5–163.5) | 3.467 | 12 | 12 |
| Eye | 139.5 (89.0–184.0) | 45.0 (30.0–67.0) | 194.0 (158.5–220.0) | 3.775 | 12 | 12 |
| Eye Vis | 171.5 (142.0–189.0) | 27.0 (22.5–34.0) | 192.5 (175.0–215.5) | 3.257 | | 12 |
| Eye Contact Lens-Sci Clin Pra | | | | 2.018 | 6 | 12 |
| Graef Arch Clin Exp Ophthalmol | 99.0 (84.5–151.5) | 13.5 (10.0–26.5) | 131.0 (104.0–162.0) | 3.117 | 12 | 12 |
| Indian J Ophthalmol | 72.5 (21.5–115.5) | 51.0 (23.5–135.5) | 134.5 (45.0–273.0) | 1.848 | 12 | 12 |
| Int J Ophthalmol | 61.5 (30.5–141.5) | 68.5 (59.0–97.0) | 145.0 (95.5–234.0) | 1.779 | 12 | 12 |
| Invest Ophthalmol Vis Sci | 108.0 (74.0–155.0) | 36.5 (30.0–72.5) | 160.0 (115.0–220.5) | 4.799 | 12 | 12 |
| Int Ophthalmol | 131.0 (108.5–146.0) | 15.0 (9.0–33.0) | 147.0 (123.5–215.5) | 2.031 | 12 | 12 |
| J AAPOS | 127.5 (77.0–166.0) | 92.5 (72.0–125.5) | 240.5 (166.0–269.0) | 1.220 | 6 | 12 |
| JAMA Opthalmol | | 60.0 (52.5–69.5) | | 7.389 | 12 | 12 |
| J Cataract Refract Surg | 159.5 (100.0–185.0) | 140.5 (120.0–166.0) | | 3.351 | 12 | 12 |
| J Eye Mov Res | | | 135.0 (100.5–151.5) | 0.957 | 6 | 12 |
| J Fr Ophthalmol | 49.0 (14.5–75.5) | 182.5 (94.5–236.5) | 254.5 (136.0–321.0) | 0.818 | 10 | 12 |
| J Glaucoma | 87.0 (56.0–136.0) | 16.5 (10.0–20.0) | 116.5 (69.0–146.0) | 2.503 | 12 | 12 |
| J Neuro-Ophthal | | | | 3.042 | 4 | 12 |
| J Ocular Pharmacol Ther | 124.0 (99.0–200.0) | 46.0 (35.5–64.5) | 183.0 (143.5–256.5) | 2.671 | 10 | 12 |
| J Ophthalmol | 125.5 (100.5–141.5) | 36.0 (17.5–43.5) | 161.5 (118.0–180.5) | 1.909 | | 12 |
| J Pediatr Ophthalmol Strabismus | 69.0 (56.0–132.0) | 34.5 (15.5–53.5) | 123.5 (77.5–173.5) | 1.402 | 6 | 12 |

| | | | | | | |
|---|---|---|---|---|---|---|
| **Table 1 (continued)** | | | | | | |
| **Journal** | **Median time between submission and acceptance (interquartile range), days** | **Median time between acceptance and publication (interquartile range), days** | **Median time between submission and publication (interquartile range), days** | **Impact factor** | **No. of issues/ year** | **No. of articles examined** |
| J Refractive Surg | 90.5 (73.0–156.0) | 4.0 (1.0–10.0) | 110.0 (78.0–158.5) | 3.573 | 12 | 12 |
| J Vision | | | 253.5 (198.5–316.0) | 2.240 | 12 | 12 |
| Jpn J Ophthalmol | 165.5 (110.5–217.5) | 63.5 (59.0–85.5) | 220.0 (166.0–277.0) | 2.447 | 6 | 12 |
| Klinische Monatsblat Augenheilkunde | 52.5 (28.0–98.5) | 79.5 (68.5–90.5) | 156.5 (100.5–177.0) | 0.700 | 12 | 12 |
| Mol Vis | 251.5 (167.0–323.0) | 2.0 (2.0–2.0) | 253.5 (169.0–325.0) | 2.367 | | 12 |
| Ocul Immunol Inflamm | 100.0 (66.0–107.5) | 42.5 (33.5–58.0) | 143.5 (95.5–169.5) | 3.070 | 8 | 12 |
| Ocul Surf | 142.5 (82.5–198.0) | 4.5 (2.5–9.5) | 147.5 (90.0–202.0) | 5.033 | 4 | 12 |
| Ophthalmic Epidemiol | 240.5 (97.0–429.0) | 18.5 (12.5–19.5) | 249.5 (113.0–450.0) | 1.648 | 6 | 12 |
| Ophthalmic Physiol Opt | 91.5 (58.5–111.0) | 38.0 (33.5–47.5) | 138.0 (104.5–150.5) | 3.117 | 6 | 12 |
| Ophthalmic Plast Reconstr Surg | | | | 1.746 | 6 | 12 |
| Ophthalmic Surg Lasers Imaging | 112.5 (74.0–175.0) | 54.5 (36.0–126.5) | 188.0 (145.0–218.5) | 1.300 | 12 | 12 |
| Ophthalmic Genet | 128.5 (80.0–208.0) | 28.5 (16.5–40.5) | 172.5 (118.5–236.5) | 1.803 | 6 | 12 |
| Ophthalmic Res | 130.5 (59.0–232.0) | 9.0 (2.0–50.5) | 155.5 (87.5–265.5) | 2.892 | 6 | 12 |
| Ophthalmologe | | | | 1.059 | 12 | 12 |
| Ophthalmologica | 113.0 (85.5–126.0) | 43.5 (25.0–82.0) | 153.0 (122.5–214.0) | 3.250 | 6 | 12 |
| Ophthalmology | 72.0 (50.5–144.0) | 7.0 (6.5–8.5) | 80.0 (57.0–151.5) | 12.079 | 12 | 12 |
| OPHTHALMOL THER | | | 58.5 (46.0–97.0) | 3.536 | 4 | 12 |
| Optom Vis Sci | 210.5 (162.5–265.5) | | | 1.973 | 12 | 12 |
| Perception | 168.5 (133.5–252.0) | 38.0 (31.5–61.0) | 230.5 (170.5–287.0) | 1.490 | 12 | 12 |
| Retin-J Retin Vitr Dis | | | | 4.256 | 12 | 12 |
| Semin Ophthalmol | 181.5 (96.0–405.5) | 21.5 (18.5–31.0) | 211.0 (119.5–449.5) | 1.975 | 6 | 12 |
| Transl Vis Sci Technol | 93.5 (54.5–123.0) | 74.0 (48.0–91.0) | 147.0 (138.5–194.5) | 3.283 | 12 | 12 |
| Vision Res | 174 (141.0–225.5) | 22.0 (14.0–25.5) | 184.5 (159.0–258.5) | 1.886 | 12 | 12 |
| Visual Neurosci | 163 (163.0–163.0) | 63.0 (63.0–63.0) | 226.0 (226.0–226.0) | 3.241 | 1 | 1 |

days. The longest time was 5.1 times longer than the shortest. The cumulative median peer review time was 118.0 (IQR, 74.0–185.0) days. The individual median times of AP ranged widely, from 2.0 to 322.0 days, while the combined median AP was 31.0 (IQR, 15.0–64.0) days. The individual median SP ranged from 58.5 to 594.5 days. The cumulative median SP was 161.0 (IQR, 111.0–232.0) days. The median impact factor of all the included journals was 2.55.

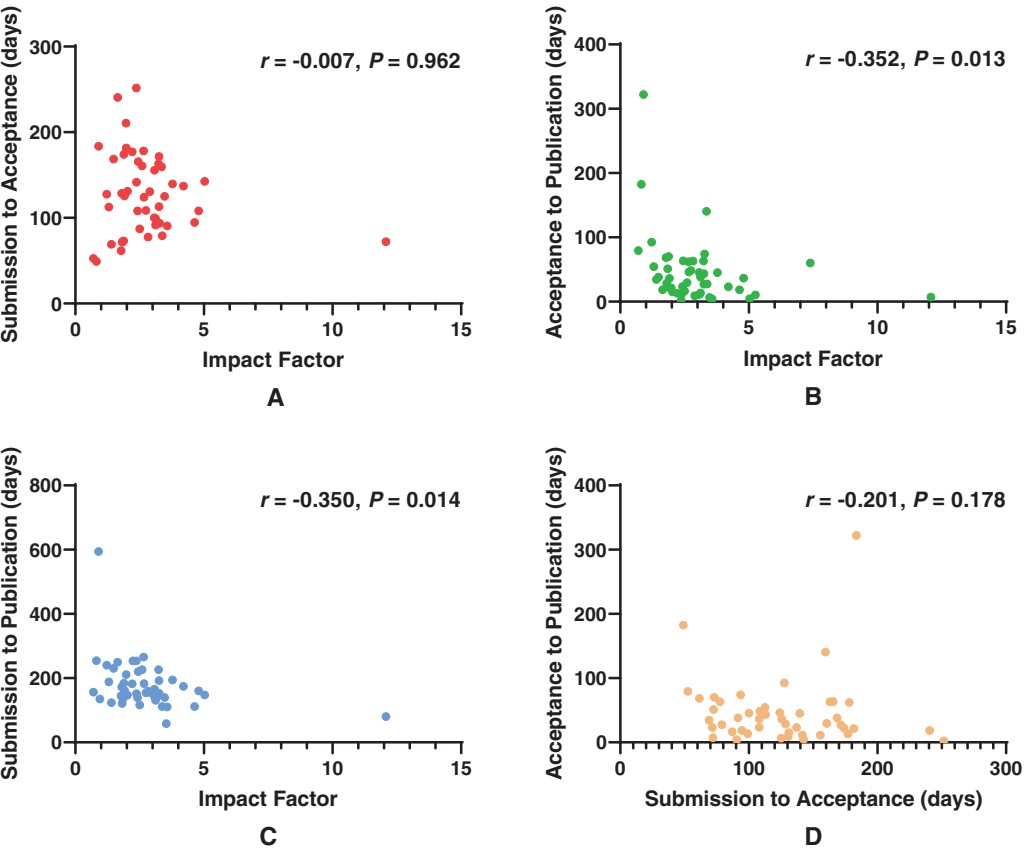

**Figure 1 Scatterplots showing correlation statistics.** (A) Correlation between the impact factor and SA ($r = -0.007$, $P = 0.962$), (B) Correlation between the impact factor and AP ($r = -0.352$, $P = 0.013$), (C) Correlation between the impact factor and SP ($r = -0.350$, $P = 0.014$). (D) Correlation between SA and AP ($r = -0.201$, $P = 0.178$).

Figure 1 shows the correlation between impact factor and publication delays of included journals. There was no correlation between the impact factor and SP ($r = -0.007$, $P = 0.962$). Negative correlations were observed between the impact factor and AP ($r = -0.352$, $P = 0.013$) and between the impact factor and SP ($r = -0.350$, $P = 0.014$). No correlation was found between SA and AP ($r = -0.201$, $P = 0.178$). After excluding the outliers, the two journals with the highest impact factors (*Ophthalmology*, impact factor: 12.079; *JAMA Ophthalmology*, impact factor: 7.389), the correlations between the impact factor and SA ($r = -0.033$, $P = 0.824$), AP ($r = -0.361$, $P = 0.013$), and SP ($r = -0.310$, $P = 0.032$) did not change significantly (shown in Fig. S1). Finally, there was no correlation between SA and AP ($r = -0.249$, $P = 0.095$) (shown in Fig. S1).

Most journals ($n = 47$; 81.03%) provided AOP. As shown in Fig. 2, the impact factors of these journals were comparable to those of journals that did not provide AOP (median, 2.651 (range, 0.700–12.079) *vs*. median, 2.240 (range, 0.957–4.799); $P = 0.677$). Moreover, as shown in Fig. 3, the median time of AP (29.0 (IQR, 15.8–57.3) days) in journals with AOP did not differ significantly from that in journals without AOP (36.5 (IQR, 27.0–63.0 days); $P = 0.606$) (shown in Fig. 3, violin plots 1 and 3). In contrast, APP was

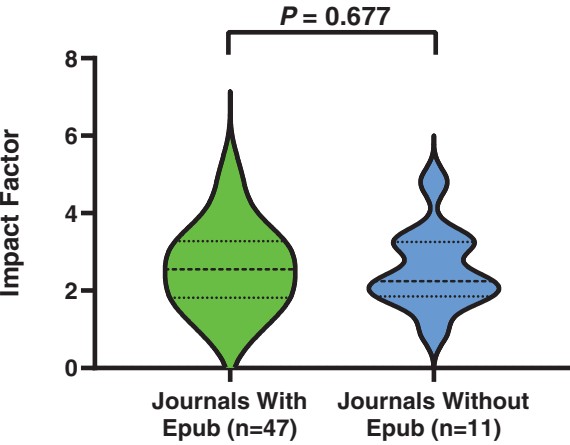

**Figure 2 The impact factor in journals with and without an advance online publication (Epub) feature.**

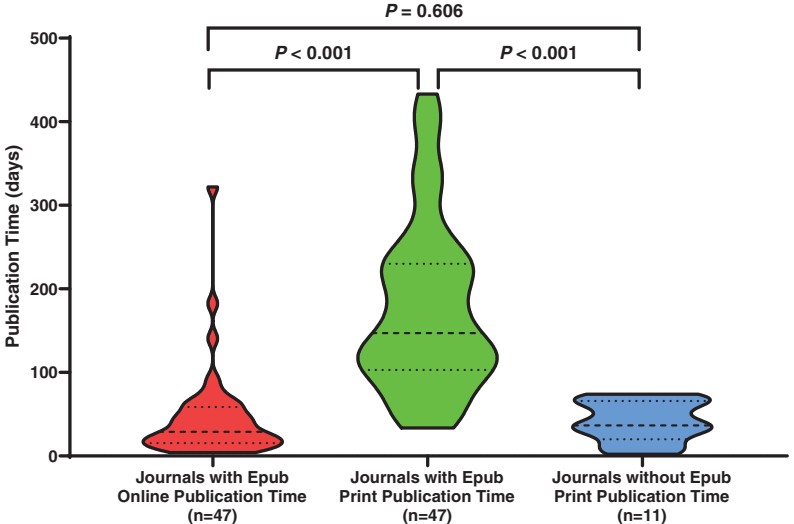

**Figure 3 Publication delays (days) in journals with and without an advance online publication (Epub) feature.**

significantly longer for journals with AOPs (147.0 (IQR, 108.0–229.5) days) than AP for journals without AOPs (36.5 (IQR, 27.0–63.0) days; $P < 0.001$) (shown in Fig. 3, violin plots 2 and 3). For journals providing AOP, the median time of AP (29.0 (IQR, 15.8–57.3) days) was significantly shorter than the median time of APP (147.0 (IQR, 108.0–229.5) days; $P < 0.001$) (shown in Fig. 3, violin plots 1 and 2).

Figure 4 shows the correlation between OA rate and publication delays of included journals. There was no correlation between OA rate and SA ($r = -0.019$, $P = 0.897$). No correlations were observed between OA rate and AP ($r = -0.112$, $P = 0.442$) and between OA rate and SP ($r = -0.198$, $P = 0.173$). Moreover, no correlation was found between OA rate and the impact factor ($r = -0.227$, $P = 0.087$).

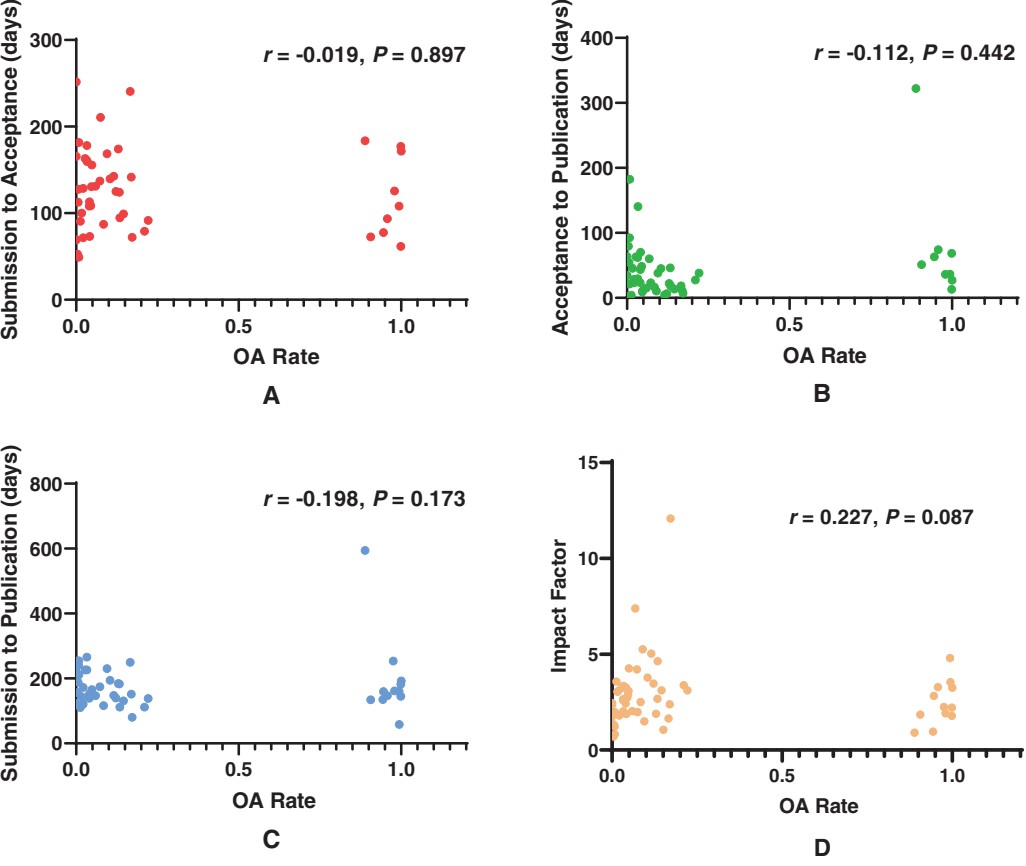

**Figure 4 Scatterplots showing correlation between OA rate and publication delays.** (A) Correlation between OA rate and SA ($r = -0.019$, $P = 0.897$), (B) Correlation between OA rate and AP ($r = -0.112$, $P = 0.442$), (C) Correlation between OA rate and SP ($r = -0.198$, $P = 0.173$). (D) Correlation between OA rate and the impact factor ($r = -0.227$, $P = 0.087$).

Figure 5 shows correlation between acceptance rates and publication delays. There was no correlation between acceptance rate and SA ($r = 0.193$, $P = 0.549$). No correlation was found between acceptance rate and AP ($r = 0.311$, $P = 0.301$). Positive correlation was observed between acceptance rate and SP ($r = 0.726$, $P = 0.007$).

## DISCUSSION

This study evaluated the article publication speeds of peer-reviewed ophthalmology journals in 2020 and explored the relationships between publication delays, impact factors, and AOP and OA status. The median times of SA, AP and SP of all ophthalmology journals were 118.0 (IQR, 74.0–185.0) days, 31.0 (IQR, 15.0–64.0) days, and 161.0 (IQR, 111.0–232.0) days, respectively. Negative correlations were observed between the impact factor and AP and SP. The acceptance rate was positively correlated with the publication delay. Besides, no correlations were found between AOP and the impact factor or AP and no correlations were observed between the OA rate and publication speed or the impact factor.

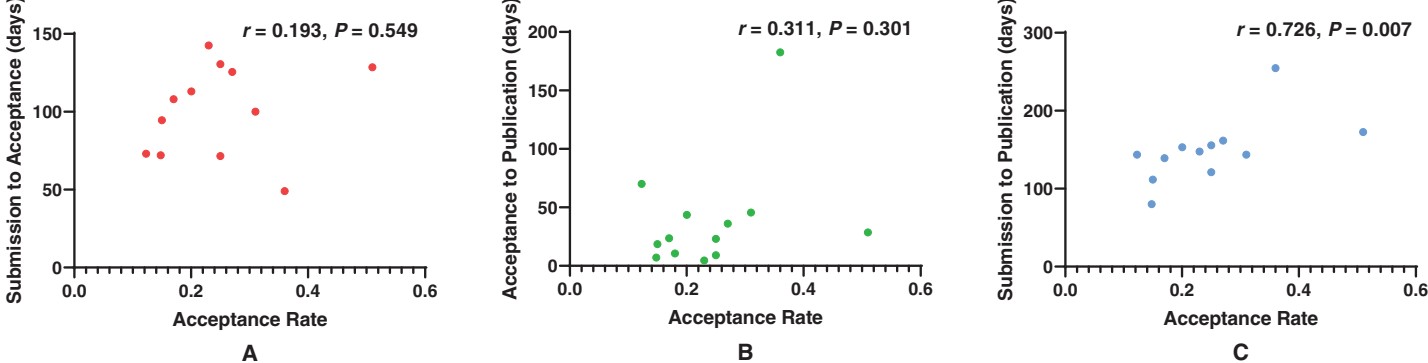

**Figure 5 Scatterplots showing correlation between acceptance rate and publication delays.** (A) Correlation between acceptance rate and SA ($r = 0.193$, $P = 0.549$), (B) Correlation between acceptance rate and AP ($r = 0.311$, $P = 0.301$), (C) Correlation between acceptance rate and SP ($r = 0.726$, $P = 0.007$).

Compared with *Chen, Chen & Jhanji (2013)* study on the publication times of 51 ophthalmology journals in 2010, the SA and AP decreased in 2020 (median SA in 2020 *vs.* 2010: 118.0 (IQR, 74.0–185.0) *vs.* 133 (IQR, 100.5–171.5) days; median AP in 2020 *vs.* 2010: 31.0 (IQR, 15.0–64.0) *vs.* 100 (IQR, 62.9–166.3) days). This may have resulted from early editorial manuscript screening that reduce the burden on reviewers (*Fernandez-Llimos, 2019*) and technological advances, such as improved peer review systems to decrease redundancy in the process (*Kelly, Sadeghieh & Adeli, 2014*) and artificial intelligence tools used in evaluating originality, validating statistics, detecting plagiarism and automatic editing manuscripts (*Baffy et al., 2020*), which have accelerated the publication process in recent years.

In this study, the individual median peer review times of ophthalmology journals varied widely (49.0–251.5 days), and the range of individual median AP was ever wider (2.0–322.0 days), with a cumulative median time of 31.0 days. The publication process can be divided into two stages: the peer review time and the time from acceptance to publication. The publication delays of ophthalmology journals may be influenced by following factors. First, the peer review stage is mainly affected by the speed at which journal editors forward the submitted articles to peer reviewers, the time it takes peer reviewers to complete the reviews, and the time it takes authors to make revisions (*Bhattacharya & Ellis, 2018*; *Björk & Solomon, 2013*). Second, peer review times may be affected by publication bias: the significance of findings may affect the publication of papers (*Song et al., 2017*). Studies have found that papers reporting positive results are more likely not only to be published but also to be published in high-impact journals and to be cited frequently (*Marín-Franch, 2018*). On the other hand, papers reporting negative results are more likely to be rejected or to undergo longer publication processes due to more rigorous reviews by editors and peer reviewers and suggestions for major revisions (*Thornton & Lee, 2000*). Publication bias can lead to overestimations of positive treatment effects in meta-analyses, inadvertently lead to selective result reports and impact reproducibility (*Marín-Franch, 2018*). To prevent publication bias, it is recommended that trials be registered beforehand and that journal editors and reviewers adopt the highest

standards in assessing studies' scientific merit and suitability for publication (*Thornton & Lee, 2000*). Although several journals are now willing to accept papers reporting negative results, more time and effort are needed to reduce publication bias (*Li, Hsueh & Liu, 2012*; *Marín-Franch, 2018*). Third, AP includes processes of copy editing, typesetting, proofreading and queuing for publication (*Björk & Solomon, 2013*), which can be accelerated by publishers' efficiency and authors' timely response. We found that 44 ophthalmology journals had higher impact factors in 2020 than in 2010 (*Chen, Chen & Jhanji, 2013*), while only three journals—*Journal of Vision*, *Molecular Vision*, and *Vision Research*—had lower impact factors compared with 2010. *Journal of Vision* and *Molecular Vision* had longer SP in 2020 (253.5 and 253.5 days, respectively) than in 2010 (216.5 and 99 days, respectively). *Vision Research* had shorter publication times than in 2010, but it had reduced its issues from 24 to 12 per year. These examples suggest that journals' publication speed and issue frequency may affect their impact factors.

We also found negative correlations between the impact factor and AP and SP. This phenomenon may be explained by that publication delays may reduce impact factors through disturbing literature citation and that high impact journals may have more efficient publication processes and more active reviewers. Based on the transfer function model of the literature citing process, research has proved an inverse relationship between a scientific field's average publication delay and journal impact factors and suggested that if a journal's publication delay increases, impact factors or journal rankings of other journals that refer to articles of this journal will decrease, and then the delay further transfer to self-citing process leading to a greater decline in the impact factor of this journal (*Yu, Guo & Yu, 2006*; *Yu, Wang & Yu, 2005*). There are also studies on the relationship between the impact factor and publication speed Which reported inconsistent results: studies on anesthesiology journals (*Mohanty et al., 2021*) and Indian biomedical journals (*Shah, Sherighar & Bhat, 2016*) have found no correlation between impact factor and publication speed. Besides, a study on otolaryngology journals (*Kalcioglu et al., 2015*) found that journals with higher impact factors took longer to accept and publish papers, perhaps due to the large number of submissions to these journals. More studies are needed to further investigate this relationship in various scientific fields.

We found no significant difference in impact factors between ophthalmology journals with and without AOPs in 2020. This differs from *Chen, Chen & Jhanji (2013)* finding that journals with AOP had statistically higher impact factors than those without this facility in 2010, which may be due to the difference in the included journals with AOPs in Chen's study compared to our current study. Compared to the journals with AOP in Chen's study, more low-impact factor journals offered AOP service in the current study. Although research reported that online-to-print delays can artificially raise a journal's impact factor, with AOP leading to earlier read and citations during the 2-year window for impact factor calculation based on print publication dates (*Tort, Targino & Amaral, 2012*), the effect of online-to-print delays in impact factors among various journals remains unknown. A study on five journals related to liver diseases including 1,039 original articles found that the low-impact factor journals had significant longer online-to-print lags than the high-impact factor journals, contrary to the hypothesis of positive association between

impact factors and the length of online-print delays (*Qi et al., 2015*). In our study, no difference in impact factors was observed in ophthalmology journals that provided AOP with an online-to-print delay than journals without AOP. Although we identified no relationship of AOP and impact factors of ophthalmology journals, AOP is a general trend in journal publishing and is considered a valuable way to shorten AP, provide convenience to authors and readers, and enable important scientific data to be disseminated rapidly, thus promoting developments in scientific research (*Amat, 2007*; *Rossor, 2005*).

In this study, the OA rate did not correlate with the impact factor in ophthalmology journals, which may be resulting from several reasons. First, the extensive web links provide easy access to articles besides OA publishing. With some authors upload their articles to subject or institutional repositories, which is called "green OA" (*Björk & Solomon, 2013*), the potential readership of subscribed articles can be expanded and the citations may increase. Second, OA is just one of several factors influencing the citation levels of particular journals, other factors including the journal prestige, the interest of article topics, the layout quality for easy reading, and timeliness of publication should also be considered (*Björk & Solomon, 2012*). Number of researchers tend to submit their papers to more established subscription journals than OA journals lacking established reputation (*Björk & Solomon, 2012*). Third, the citation advantage offered by the OA model remains controversial. Some studies reported that OA articles attract more citation than non-OA articles (*Gargouri et al., 2010*; *Norris, Oppenheim & Rowland, 2008*), while others found no difference in citations between OA and non-OA journals (*Narayan, Lobner & Fritz, 2018*). The OA citation advantage may mainly result from high-quality articles that readers prefer to cite rather than authors' selections to make OA, as the top 20% of articles receive about 80% of all citations (*Norris, Oppenheim & Rowland, 2008*). OA maximizes accessibility of high-quality articles and thereby enhances citations (*Gargouri et al., 2010*). Findings on the relationship between OA and impact factors are also inconsistent. A study on oncology journals (*Hua et al., 2017*) reported that OA was associated with more citations and higher impact factors. Conversely, a study on orthopedic journals (*Sabharwal, Patel & Johal, 2014*) found no differences between fully OA and hybrid OA journals. Likewise, we found the OA rate did not correlate with impact factors in ophthalmology journals.

There is no correlation between publication delays and OA rates, contrary to the popular belief that OA journals have faster publication times, which may because of the small size of ophthalmology journals with high OA rate. Research on the relationship between OA and publication speed is scarce. *Björk & Solomon (2013)* found that review and publication delays tend to be shorter for OA journals, and that original OA journals rather than that were converted from subscription appear to publish articles more quickly than subscription journals. However, authors considered that this finding should be interpreted with a great deal of caution given the small number of OA journals in the study (*Björk & Solomon, 2013*). In our study, OA rates are not evenly distributed and only 20.69% (12/58) journals had an OA rate over 23%, which may be partly explained the different finding with the prior study.

The positive correlation of the acceptance rate and SP may be partly explained by journal impact. For high-impact journals with a lower acceptance rate, most unsuitable manuscripts could be quickly rejected during the editors' screening, which can reduce burdens of reviewers (*Björk & Solomon, 2013*). Moreover, high impact journals might find it easier to recruit reviewers and appear to be more efficient both in acceptance and publication processes (*Björk & Solomon, 2013*).

Certain limitations of this study should also be mentioned. First, although publication dates of most included articles were available, there were seven (12.1%) and six (10.3%) journals did not indicate submission and acceptance dates, respectively, leading to missing data of publication delays that may influence the evaluation of publication speed in ophthalmology journals. Second, we did not evaluate the impact of other bibliometric indicators, such as immediate indexing and cited half-life, nor did we assess the impacts of study designs and reported results on the publication speed. Third, we analyzed data only from 2020, which did not allow an investigation of longitudinal trends in publication delays. Future research focus on longitudinal trends in ophthalmology journals are warranted.

In conclusion, our findings show that publication delays of ophthalmology journals have been shortened in 2020 than that in 2010, indicating the overall peer review and publication processes have been accelerated. Our results also indicate negative correlations between impact factors and AP and SP, which may be the result of the publication delay effect on impact factors and efficient peer review processes in high impact journals. Journals with a lower acceptance rate are more likely to have shorter publication delays. AOP and OA seem to make no difference in the impact factors or publication speeds of ophthalmology journals.

## ABBREVIATIONS AND ACRONYMS

| | |
|---|---|
| **AOP** | Advance online publication |
| **AP** | The times from acceptance to online publication of journals with advance online publication or print publication of that without advance online publication |
| **APP** | The times from acceptance to print publication of journals with advance online publication |
| **IQRs** | Interquartile ranges |
| **OA** | Open access |
| **SA** | The times from submission to acceptance |
| **SP** | The times from submission to online publication of journals with advance online publication or print publication of that without advance online publication |

### Funding

This work was supported by the National Natural Science Foundation of China (81873675). The funders had no role in study design, data collection and analysis, decision to publish, or preparation of the manuscript.

### Grant Disclosures

The following grant information was disclosed by the authors:
National Natural Science Foundation of China: 81873675.

### Competing Interests

The authors declare that they have no competing interests.

### Author Contributions

- Yinglin Yu performed the experiments, prepared figures and/or tables, and approved the final draft.
- Wei Li performed the experiments, prepared figures and/or tables, and approved the final draft.
- Chaoqun Xu performed the experiments, authored or reviewed drafts of the article, and approved the final draft.
- Yuan Tan analyzed the data, authored or reviewed drafts of the article, and approved the final draft.
- Weining Zhu analyzed the data, authored or reviewed drafts of the article, and approved the final draft.
- Bowen Zhang analyzed the data, authored or reviewed drafts of the article, and approved the final draft.
- Yingshi Zou analyzed the data, authored or reviewed drafts of the article, and approved the final draft.
- Leyi Hu analyzed the data, authored or reviewed drafts of the article, and approved the final draft.
- Guangming Jin conceived and designed the experiments, prepared figures and/or tables, and approved the final draft.
- Zhenzhen Liu conceived and designed the experiments, prepared figures and/or tables, and approved the final draft.

### Data Availability

  The raw data are available in the Supplemental File.

### Supplemental Information

Supplemental information for this article can be found online at http://dx.doi.org/10.7717/peerj.14331#supplemental-information.

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
