# Peer review of "Publication delays and associated factors in ophthalmology journals"

_PeerJ, doi:10.7717/peerj.14331_

## Round 0.1 · original submission · Major Revisions

Both reviewers have made some excellent suggestions which should improve the paper significantly. Therefore, I invite you to respond to the reviewers' comments and revise your manuscript.

·

Basic reporting

First of all, I would like to congratulate the authors for their work. The manuscript is well structured, presents relevant results and it is well written. I have some comments and suggestions for the consideration of the authors.

Consider changing the title: the study is not comprehensive as only one article per issue of each journal was explored, neither updated, unless there is previous research by the same team exploring these same aspects. The authors correctly note that previous research was conducted employing different methods and sample sizes. Therefore, I would suggest removing the term updated from the title.

Introduction, line 75: Although it may be true that OA has contributed to removing the barriers to access scientific knowledge, may journals offering OA options require authors to pay high publication / processing fees to do so, which may hinder the access to publication in this format to researchers and institutions without funding.

References, tables, figures, and data are adequate.

Experimental design

Methods, line 110: Some journals without print (paper) copies of their issues also publish ahead-of-print articles. I would suggest adding as a definition of “print publication date” the date in which the article was actually printed or the date in which final pagination / bibliometric details were added to the article.

Methods: I am a bit concerned about the possible difference in the number of articles explored for each journal. As the authors point out, not all journals publish a new issue each month. For those journals with less frequent publication, only one article was chosen (this reduced the number of articles to 4 in some cases). The authors may have considered selecting 12 articles at random for each journal, one per month for those publishing monthly and N per issue for those publishing with less frequency, where N x n. issues = 12.

Methods: Given that JCR currently lists the percentage of OA for each journal within the category Ophthalmology, it may be an interesting addition to the manuscript to include this information, rather than a dichotomous OA / NON OA. This would allow to conduct more insightful quantitative analysis, such as exploring possible correlations between % of OA and publications times.

Methods are described in detail and the correct analysis is conducted.

Validity of the findings

Methods: Another interesting addition would be to know the % of acceptance for each journal. This information is commonly published in the webpage of journals so that authors may have an idea of the probability of receiving a rejection letter for their manuscript. It may be that journals with a higher rejection rate then publish faster those articles that have been accepted. In the same line, it would also be interesting to know whether publication times vary depending on the total number of published articles per year in each particular journal.

Findings are relevant and of interest to both clinical and bibliometric researchers.

Additional comments

References: Please make sure that all references follow the same format. For instance, format of DOI or format of CAPS at the First Letter of Each Word of the Title of the Article.

Reviewer 2 ·

Basic reporting

The authors constantly use term „publication lags” instead of, for example: publication delay. For solely language pietism, application of synonyms would be helpful.
Line 76-78: citation/’s needed
Authors persistently repeat phrases „median time/’s form submission to publication”, „Median time/’s from acceptance to advance online publication” etc. For manuscript clarity, I recommend use of predefined abbreviation for example (AO: time from acceptance to advance online publication). It would enhance manuscript overall read-through experience.
Line 143: Word „collective” was inappropriately used. I suggest „cumulative”.
Line 250: Referencing to ecology journal and supporting findings with this article in manuscript mounted in medical field is inappropriate. Different research importance, quality, value and influence between these fields.
Raw data shared is professionally formatted, in xlsx format accessible my most of the readers. The article is well-structured, follows the scientific rigor with results that support the research question. English is clear and easy to understand by international audience, however some improvements (suggested above) should be done.

Experimental design

Lines 97-98: Authors did not state what type of article they include as the original study. For example, some journals count review articles in non-review-only journals as a original articles.

Lines 120-122: Authors did not mention, why had they decided to use non-parametric tests instead of those parametric statistical analysis. Had they performed a Shapiro- Wilk test (included in R-package) before test selection?

Lines 153-158: I do not understand why authors excluded top cited journals. It seems like they were searching for any statistically significant results. Impact Factor is relatively constant value. Data extracted from small number of articles in each journal could potentially influenced statistical significance.

Finally, research is within Aims and Scope of the journal. Hypothesis was well defined, identified knowledge gap is some of interest, however calculation of publication times in some journals duplicate previous literature. Methods are sufficient to replicate the study.

Validity of the findings

Lines 189-191 and Lines 266-277: Advance online publication gives article more visibility, thus increase its potential to get a citation. This is common trick used by publishers to robust Impact Factor. Impact Factor is average number of citations per article counted in 2-year period from in-print publication. Advance online publication gives extra time for article to be cited. No statistical correlation between advance online publication and the impact factor is curious finding.

Authors’ comment and support for this conclusion is scare. In my firm believe, they should describe how included journals have changed advance online publishing policy, since 2013. Chen et al. (DOI: 10.1016/j.ophtha.2013.01.044) explicitly stated that journals with advance online publication had higher impact factors compared with those without it. Suggesting that increased popularity of advance online publication may be a reason is insufficient.

Lines 192-193 and lines 289-301: No difference between OA and traditional journals sounds to be interesting. OA journals are accessible for everyone. Traditional requires subscription, which is often costly and limited. In context of higher visibility, OA journals should have had statistically more citations. However, what is the rationale behind this finding? Perhaps, top journals have better selection process and publish articles that are more interesting for the scientific community. This rationale should be stated in discussion. Introducing findings from different not related disciplines (fuel journals!) aimed at avoiding intellectual discussion, simplified the discussion to „inconclusive results”, and spared the authors effort.

Lines 302-310: A few articles in each journal were included in the analysis. The investigation could be more contributing if the authors had analyzed more articles (for example random 10 original studies for each issue). Total number of analyzed articles compared to previous literature is relatively small. There is great disproportion between analyzed journals in number of included studies. Previous literature (Chen et al, 2013) involved the same number of articles for each journal. This inequality could significantly influence demonstrated results. This is the most important limitation of the study, not other bibliometric analysis etc.

---

## Round 0.2 · accepted · Accept

I am pleased to inform you that the one reviewer who submitted a review of the revised manuscript recommended acceptance of your manuscript for publication. Congratulations.

·

Basic reporting

No comment

Experimental design

No comment

Validity of the findings

No comment

Additional comments

Thank you for uploading a revised version of this manuscript addressing previous comments and suggestions. I have no further comments.